# A Regulatory-Compliant Genotoxicity Study of a Mixture of C60 and C70 Fullerenes Dissolved in Olive Oil Using the Mammalian Micronucleus Test

**DOI:** 10.3390/nano15110870

**Published:** 2025-06-05

**Authors:** Fathi Moussa

**Affiliations:** Institut de Chimie Physique, CNRS–UMR 8000, Paris–Saclay University, 91400 Orasay, France; fathi.moussa@universite-paris-saclay.fr

**Keywords:** fullerenes, C60, C70, oil, genotoxicity, regulation

## Abstract

Although they show great promise in the medical field, the safety of fullerenes—discovered forty years ago—is still uncertain, according to regulatory experts at the European Scientific Committee on Consumer Safety. This is a major obstacle to progress in the field. Meanwhile, oily solutions of fullerenes intended for human and pet consumption can be purchased online, without any marketing authorization. Therefore, to avoid any potential public health issues, regulatory-compliant preclinical studies on fullerene oily solutions are urgently needed. We present the first in vivo genotoxicity study of a C60/C70 fullerene mixture (4.1/1, *w*/*w*) dissolved in extra virgin olive oil (0.8 mg/mL). The study was conducted using the Mammalian Micronucleus Test (MMT) in an independent GLP-laboratory, in compliance with the OECD and EPA guidelines. The MMT was performed on NMRI mice following the oral administration of fullerenes at doses of up to 3.6 mg/kg. This dose is almost the maximum dose that can be administered to rodents. The data obtained clearly show that fullerene oily solutions have no genotoxic activity under these conditions. This should pave the way for further regulatory investigations of fullerene oily solutions.

## 1. Introduction

Fullerenes (Figure 1) were discovered forty years ago [1]. Despite their enormous potential for biomedical applications, recognized since 1996 [2], no such applications have materialized to date [3,4,5]. This is mainly due to doubts about their safety, as reflected in a recent report by experts from the European Commission’s Scientific Committee on Consumer Safety [6]. Although the safety of pure C60 fullerene (C60) has been confirmed by independent research teams [7,8,9], conflicting data continues to cast doubt [10,11,12,13,14,15,16,17,18,19,20,21,22]. This is because fullerene particles may contain residual impurities, including toxic elements [7,8,9]. This means that impure fullerene materials can be highly toxic, as has been demonstrated since 2007 [7,8,9].

Nevertheless, fullerenes are showing great promise in many areas of healthcare, as evidenced by the growing number of publications on the subject; there are too many to mention exhaustively. Here, we will only quote the most recent ones [23,24,25,26,27,28,29,30,31,32,33,34,35,36,37,38,39,40,41] and some of the most recent general reviews [42,43,44,45,46,47].

The main problem nowadays is that, since a 2012 paper showed that C60 dissolved in extra-virgin olive oil (EVOO) could extend the lifespan of rats in an experimental model [19], the product has been marketed by countless websites online. However, there is a lack of standardization in dosing [43]. Most importantly this product has not been approved by any major health authorities, such as the FDA, MHRA or EMA [43]. The risk is therefore twofold. Firstly, of course, there is the risk to consumers’ health. Secondly, even the slightest incident could permanently damage the reputation of this promising material. Therefore, preclinical trials that are conducted in accordance with regulatory guidelines and Good Laboratory Practice (GLP) are urgently needed in order to address concerns about the safety of fullerene oily solutions [4]. In this context, a recent regulatory study conducted in an independent GLP-accredited laboratory showed that pure C60 dissolved in extra virgin olive oil (EVOO) has no acute toxicity after daily oral administration to rodents [5]. However, to the best of our knowledge, a certified report on the genotoxicity of fullerenes dissolved in EVOO is still lacking.

Since we showed that fullerene particles suspended in a biocompatible solvent can be administered to rodents up to 5.0 g/kg body weight (bw) [48], many academic studies have been devoted to the study of the genotoxicity of fullerene particle suspensions [13,14,15,16,17,18,20]. However, it is unlikely that this type of suspension would be ingested intentionally. By contrast, there are numerous videos online of people drinking fullerene oily solutions.

Here, we report the results of the first regulatory-compliant in vivo genotoxicity study of a mixture (4.1/1, *w*/*w*) of the two most abundant fullerenes, C60 and C70 (Figure 1), dissolved in EVOO, conducted in an accredited GLP laboratory. In accordance with OECD [49] and EPA [50] guidelines, this study uses the in vivo mammalian micronucleus test (MMT) in a mouse experimental model.

According to the manufacturers, the C60/C70 mixture used in this study is a more practical industrial composition than a pure C60 solution. This is particularly pertinent given that it avoids the need to separate these the two most abundant fullerenes [51]. Furthermore, as has already been demonstrated, C70 and C60/C70 mixtures exhibit interesting antioxidant properties [52].

This study aims to determine whether oily solutions of fullerenes, such as those sold online as dietary supplements, have genotoxic effects that result in the formation of micronuclei (MNI) in the erythrocytes of treated animals. The in vivo MMT is used to detect substance-induced damage to the chromosomes or mitotic apparatus of erythroblasts by analyzing erythrocytes collected from the bone marrow of animals, usually rodents [49,50].

The purpose of the MMT is to identify substances that cause cytogenetic damage, resulting in the formation of MNI containing lagging chromosome fragments or whole chromosomes. When a bone marrow erythroblast develops into a polychromatic erythrocyte (PCE), the main nucleus is extruded. Any micronucleus that forms may remain behind in the otherwise anucleated cytoplasm. An increase in the frequency of micronucleated polychromatic erythrocytes (MPCE) in treated animals is indicative of induced chromosomal damage (i.e., genotoxicity) [49,50].

The Appendix A, which consists of the original report as provided by the accredited European laboratory that conducted the entire study, includes the legal certifications and raw data of this study.

## 2. Materials and Methods

The Hungarian Regulation on Good Laboratory Practice Regulation: 42/2014 (VIII. 19.) EMMI Decree of the Minister of Human Capacities corresponding to the OECD GLP, ENV/MC/CHEM(98)17 [OECD Principles of GLP as revised in 1997, published in ENV/MC/CHEM (98)17); OECD, Paris, 1998] (Appendix A) approved all experimental protocols. The study was conducted according to GLP as required by the sponsor for regulatory purposes. Therefore, except for the formulation analysis, the principles of GLP according to Hungarian and international legislation were followed.

This study was performed in compliance with the procedures indicated by the following internationally accepted guidelines and recommendations: Ninth Addendum to OECD Guidelines for Testing of Chemicals, Section 4, No. 474, “Mammalian Erythrocyte Micronucleus Test.” adopted 29 July 2016 [49] and EPA Health Effects Test Guidelines, OPPTS 870.5395 “Mammalian Erythrocyte Micronucleus Test” August 1998 [50].

The Standard Operating Procedures (SOPs) detail all procedures described in the study plan, in the SOP manuals of the operational departments of Toxi-Coop Zrt. The study plan and the different phases of the study were reviewed by the QA department. The final report was reviewed according to the internal SOPs.

The Institutional Animal Care and Use Committee (IACUC) of Toxi-Coop Zrt. approved the study and signed the study plan (SOP: ALT 023-Animal Care Instructions). The entire study was conducted in compliance with the principles of Hungarian Act 2011 CLVIII (amendment of Hungarian Act 1998 XXVIII) and Government Decree 40/2013 on the Protection of Animals and in accordance with the National Research Council Guide for the Care and Use of Laboratory Animals. All methods are reported in accordance with ARRIVE guidelines.

All documents and samples related to the study are archived according to OECD GLP and Toxi-Coop Zrt. SOPs in the archives of Toxi-Coop Zrt. (Berlini utca 47-49. H-1045 Budapest, Hungary), including the Study Plan (15 years), one sample of the test item for 5 years, blocks and slides of organs and tissues (12 years), correspondence (15 years), full raw data (15 years), and one original Final Report for 15 years.

### 2.1. Experimental Design

#### 2.1.1. Test Item and Chemicals

##### Test Item

The mixture of C60 and C70 (4.1/1, *w*/*w*, purity 99.98%) dissolved in EVOO (0.8 mg/mL, corresponding to 0.09% *w*/*w* of the final solution, given a density of 893 mg/mL for the test item), trademarked as Ageon™, was prepared by SES Research Inc. (Houston, TX, USA) (Appendix A). The physicochemical characterization of the C60 used (purity: 99.98%) was performed previously, both in the solid state and in solution [4]. The physicochemical characterization of C70 was also carried out by various analytical methods, as previously described [49]. Finally, the proportions of the 2 compounds in the test item were confirmed by HPLC (Appendix A).

EVOO was chosen as the solvent for the mixture of fullerenes because of its nutritional properties and specifically for its biocompatibility and fullerene solubility [5]. The EVOO-C60/C70 solution (test item) was prepared as previously described for C60 [5], with some modifications. Briefly, 0.09 g of a C60/C70 mixture (85/15, *w*/*w*) was added to 100 mL of olive oil. After stirring for 2 weeks in the dark, the mixture was centrifuged at 5000× *g* for 1 h at room temperature and the supernatant was filtered through a Millipore filter with a pore size of 0.25 μm. The resulting solution is a transparent brown-reddish liquid with a faint oily odour containing 800 mg of fullerenes (C60/C70, 4.1/1, *w*/*w*) per litre of EVOO, as determined by liquid chromatography (Appendix A). This corresponds to 0.0896% (*w*/*w*) of fullerenes in olive oil, giving an average density of extra virgin olive oil containing the fullerene mixture of 0.893 g/mL.

A preliminary non-GLP solubility study was performed on the test item in Helianthii Annui Oleum Raffinatum (HAOR) (MAGILAB KFT, Budapest, Hungary). A homogeneous formulation was obtained in this solvent up to a concentration of 200 mg/mL, which corresponds to 0.18 mg of fullerenes per ml. For treatment, the test item was dissolved in HAOR. After weighing the required amount of test element in a calibrated volumetric flask, a partial volume of HAOR was added and the formulation was stirred until a homogeneous solution was obtained. Three concentrations, 50 mg/mL, 100 mg/mL, and 200 mg/mL of the test item, were used for the treatment. The formulations were prepared extemporaneously on the day of administration and used within 15 min without further analysis.

##### Controls

We used the solvent alone (HAOR) as a negative control and cyclophosphamide (Sigma-Aldrich St. Louis, MO, USA) dissolved (6 mg/mL) in aqua ad injectabilia for treatment as a positive control.

##### Chemicals

The other chemicals used in this study were fetal bovine serum (Sigma-Aldrich St. Louis, MO, USA), Giemsa stain (Merck KGaA, Darmstadt, Germany), aqua purificata (Magilab KFT, Hungary), methanol (Lach-Ner, Neratovice, Czech Republic), and E-Z MountTM (Epredia, Enhancing Precision Cancer Diagnostics, Kalamazoo, MI, USA).

#### 2.1.2. Test Animals

The NMRI mouse (Win: NMRI mice, TOXI-COOP ZRT, Budapest, Hungary) was selected for the study because it is a standard animal used internationally for this type of mutagenicity testing. It has been shown since 1996 that C60 can be administered intraperitoneally as a suspension to Swiss mice at doses of up to 5.0 g/kg bw without acute or subacute toxicity [48]. In this solid form, C60 accumulates selectively in the reticuloendothelial system, mainly in the liver and spleen [46].

##### Number and Groups

For a preliminary toxicity test, 2 groups of 2 male and 2 female mice per group were used. For the main study, we used 25 + 2 males randomly divided into 5 groups (5 males/group and 7 males in the high dose group). At the start of treatment, all animals were free of specific pathogens and in acceptable health condition. They were 8 weeks old and weighed 32.2 to 37.8 g, as recorded for all animals at randomization and before dosing.

##### Identification and Randomization

Individual animals were identified by numbers on the tail. Cages were marked with identification cards containing information on cage number, study code, sex, dose group, and individual animal numbers, as well as species and strain of animal. Animals were randomly assigned to the control and test groups using a randomization scheme. Randomization was verified using actual body weights to check for homogeneity and between-group variation.

##### Husbandry

For a 6-day pre-treatment acclimation period and during treatment, mice were housed in polypropylene/polycarbonate type I cages (2 animals/cage in the preliminary and high-dose groups of the main study and 5 animals/cage in the other groups of the main study) and maintained in an air-conditioned room with 12 h of artificial light daily from 6:00 a.m. to 6:00 p.m. Temperature (22 ± 3 °C) and relative humidity (40 to 70%) were recorded daily during the study. Cages were labelled with identification cards containing the study number, control or test name, group number, test serial number, sex, cage number and individual animal numbers, treatment start date, and sacrifice date. Rodents were housed in groups to allow for social interaction and with deep wood sawdust bedding to allow for digging and other normal rodent activities.

The animals were fed ad libitum with a pellet diet (ssniff^®^ SR/M-Z+H) manufactured by ssniff Spezialdiäten GmbH (Experimental Animal Diets Inc., Soest, Germany) (Appendix A). Consumption was visually monitored on a daily basis. Water was provided ad libitum from 250 mL bottles, as for human consumption.

### 2.2. Experimental Procedure

A preliminary toxicity study was conducted to determine the appropriate Maximum Tolerated Dose (MTD) for the main study, as no mouse toxicity data were available for soluble fullerenes at the time this study was initiated. The treatment was performed in HAOR solvent at a constant volume (10 mL/kg body weight), which is the maximum volume to be administered under these conditions according to OECD and EPA recommendations [49,50]. Therefore, 2000 mg/kg bw of the test substance was administered by gavage twice 24 h apart to two groups of males and two groups of females. The animals were observed regularly for signs of toxicity and mortality.

Since no deaths or signs of toxicity were observed in either sex, the doses of 500, 1000, and 2000 mg/kg bw were selected for the main test using only male mice.

#### 2.2.1. Micronucleus Test

The main study was conducted in male mice because the toxic effect of the test substance in the preliminary acute oral toxicity study was similar in both sexes.

Mice were randomly divided into five groups to receive 500, 1000, or 2000 mg/kg bw, solvent only (negative control), or 60 mg/kg bw cyclophosphamide dissolved in water for injection (positive control). Two additional male mice were dosed in the highest test item treated group to replace any which died before the scheduled sacrifice time.

After a six-day acclimation period, the test/solvent was administered orally by gavage twice at 24 h intervals, while cyclophosphamide (positive control) was administered intraperitoneally. The treatment volume was 10 mL/kg body weight. Sampling was performed once at 24 h after the second treatment, and five male animals were used for sampling. Mice were observed for visible signs of treatment response immediately after dosing and periodically until sacrifice. Sampling was performed once at 24 h after the second treatment.

#### 2.2.2. Bone Marrow Preparation and Staining

Bone marrow was harvested from two exposed mouse femurs immediately after sacrifice (cervical dislocation) at each time point. Bone marrow was rinsed with fetal bovine serum (5 mL). After vortex mixing, the cell suspension was concentrated by centrifugation and the supernatant was discarded. The cell pellet was spotted onto standard microscope slides. The slides were then dried at room temperature. The slides were then fixed in methanol for at least 5 min, air dried, and stained with Giemsa solution (10%) for 25 min. Stained slides were coated with Micromount after rinsing in distilled water and drying at room temperature for at least 12 h.

#### 2.2.3. Examination of Slides

Prior to microscopic analysis, one slide from each animal was assigned a code number for blind microscopic analysis. The code labels covered the original animal numbers to ensure that the slides were scored without bias. Four thousand polychromatic erythrocytes (PCEs) were scored per animal to evaluate micro-nucleated cells. The frequency of micro-nucleated cells was expressed as a percentage of micro-nucleated cells based on the first 4000 PCEs counted in the optical field. The proportion of immature erythrocytes to total (immature + mature) erythrocytes was determined for each animal by counting a total of at least 500 erythrocytes.

Micronucleated erythrocytes are identified using classical micronucleus criteria. These criteria consist of the presence of a uniform, brightly stained, blue-violet, round particle of a specific size within the cell. Under the microscope, the particle should appear uniform in colour, refraction and shape. Cells containing two or more micronuclei are counted as a single micronucleated cell.

The toxicologist leading the study deemed the MNT to be acceptable if the negative control data were deemed suitable for inclusion in the laboratory’s historical control database. He also required positive and scoring control results demonstrating a statistically significant increase compared to the negative control. These results also had to align with those in the historical database. Finally, he stipulated that a sufficient number of doses and cells must be analyzed. Therefore, at least five animals were used in both the treated and control groups at each sampling time point.

### 2.3. Statistics and Evaluation of Experimental Data and Interpretation

Statistical analysis was performed using SPSS Statistics v27 PC+ software for the following data: the frequencies of micro-nucleated polychromatic erythrocytes in the animals of the test and positive groups were compared with those in the corresponding negative and historical control groups; the percentage of immature erythrocytes out of the total (immature + mature) erythrocytes in the test and positive control groups were compared to those in the corresponding negative and historical negative and historical control groups. Data were analyzed for a linear trend in mutant frequency versus treatment dose using regression analysis using Microsoft Excel software.

The toxicologist in charge will consider the test to be conclusive and positive if all the acceptability criteria are met and a statistically significant increase in the frequency of micronucleated immature erythrocytes is observed in at least one treatment group compared to the negative control group. The increase must be dose-related at one sampling time, as evaluated by an appropriate assay. The result must also fall outside the range of historical negative control data (e.g., 95% Poisson-based control limits) (Appendix A).

The study manager will deem the test to be conclusively negative if all the acceptability criteria are met, and if no statistically significant increase in the frequency of micronucleated immature erythrocytes is observed in any of the treatment groups compared to the negative control group. He also stipulates that there must be no dose-dependent increase at any time during the sampling process. Finally, he states that, when evaluated using an appropriate test (e.g., 95% Poisson-based control limits), all results must fall within the distribution of historical negative control data. These conditions can only be considered if the sample being tested has come into contact with bone marrow. It should be noted that no deviations from the protocol or test guidelines were observed.

## 3. Results

### 3.1. Preliminary Toxicity Test

After two treatments, 24 h apart, at a dose of 2000 mg/kg bw of the test item, containing 3.6 mg/kg bw fullerenes, by gavage, the animals were observed periodically for signs of toxicity and mortality. No deaths or signs of toxicity were observed in either sex; therefore, doses of 500, 1000, and 2000 mg/kg bw were selected for the main study.

### 3.2. MMT

Five groups of five randomly selected mice received 500 (low), 1000 (medium), 2000 mg/kg bw (high), solvent only (negative control), or 60 mg/kg bw cyclophosphamide dissolved in water for injection (positive control) (Table 1).

Test items and solvent were administered orally by gavage twice 24 h apart, and sampling was performed once 24 h after the second treatment. Cyclophosphamide was administered intraperitoneally with a treatment volume of 10 mL/kg body weight.

Mice were observed regularly for visible signs of reactions to treatment immediately after dosing and periodically until sacrifice. No animals died during the study and no adverse reactions were observed.

Twenty-five animals were used for sampling. Bone marrow was harvested from two exposed mouse femurs immediately after sacrifice at each time point. After preparation and staining with 10% Giemsa (Appendix A), one slide from each animal was assigned a code number for blinded microscopic analysis. Because no death occurred in the original population (five animals), bone marrow smears were not prepared from the two additional mice dosed in the highest (2000 mg/kg body weight) test item.

### 3.3. Frequency of Micronuclei

The proportion of immature erythrocytes to total erythrocytes was determined for each animal by counting a total of at least 500 erythrocytes. Table 2 summarizes the frequency of MPCs, expressed as a percentage of MPCs based on the first 4000 PCEs counted in the optic field.

Cyclophosphamide-treated mice (60 mg/kg bw) showed a large statistically significant increase in the number of MPCEs compared to negative and historical controls, validating the study according to OCDE [49] and EPA [50] guidelines.

While the number of PCEs 24 h after the second treatment was not affected in the 500 mg/kg bw dose group, the reduction in the PCE/total erythrocyte ratio in the 2000 mg/kg bw dose group (Table 2) is considered biologically significant. This usually indicates that the test substance or its degradation products effectively reached the bone marrow and caused toxicity there, which also validates the study according to OCDE [49] and EPA [50] guidelines.

Oral administration twice at 24 h intervals of 500, 1000 mg/kg, or 2000 mg/kg bw of the test substance did not induce an increase in the frequency of MPCEs in male mice 24 h after the second treatment compared with the concurrent negative (solvent) and historical control groups. Therefore, the fullerene oil solution showed no genotoxic activity in the mouse micronucleus test.

## 4. Discussion

Pioneering toxicity studies have shown that it is possible to administer high doses of C60, up to 5.0 g/kg, to mice without evidence of acute or subacute toxicity [48]. It was sufficient to disperse C60 nanoparticles in a dispersant (e.g., Tween 80) and stabilize the suspension by increasing the viscosity of the medium by adding carboxymethylcellulose (CMC) [46]. The lack of toxicity up to 2.5 g/kg bw was confirmed in 2004 when this preparation was used in rats to demonstrate that C60 is a powerful antioxidant in vivo and to provide pharmacokinetic data [10]. This type of formulation has only been developed for peritoneal administration, or for intravenous administration provided that the concentration is drastically reduced and the particle size is carefully controlled to avoid microvascular occlusion.

The first in vivo genotoxicity study using C60 particles suspended in water was conducted in 2006 [13]. This study demonstrated an absence of toxicity or genotoxicity in well-established in vitro experimental models, and following the oral administration of doses of up to 2000 mg/kg bw to rats [13], thus confirming earlier results. However, subsequent publications are contradictory [11,12,14,15,16,17,18], including studies involving fullerene particles suspended in corn oil at a dose of 1000 mg/kg bw [53]. This is probably because some fullerene particles may contain residual impurities [7,8,9]. In any case, this is why the SCCS has raised concerns about the safety of fullerenes [6].

The results reported here differ significantly because they target the test item offered to consumers online, rather than the C60 particles themselves. For this reason, we tested the genotoxicity of the fullerene/olive oil mixture (the test item) using HAOR as the solvent rather than EVOO. The obtained data clearly show that the fullerene/olive oil mixture did not exhibit any genotoxic activity in the MMT, under these conditions.

Although fullerenes are only soluble in extra virgin olive oil (EVOO) at a concentration of around 0.8 mg/mL, experimental studies have shown that C60 becomes approximately 100 times more active in a living organism when it is dissolved than when it is present as particles [19]. It also becomes active immediately [10,19]. This is because C60 is only active in soluble form, i.e., when its conjugated double bonds are accessible [10,19]. C60 particles, however, are very difficult to dissolve, particularly in biological media [10,19].

The dose of fullerenes dissolved in EVOO used in this study, at 3.6 mg/kg bw, is clearly very low compared to the doses usually tested for fullerene particles. However, this dose is very close to the maximum allowed for rodents when considering the solubility in olive oil [5]. It is also very close to the most effective dose against oxidative stress in an experimental model in rats [19]. Furthermore, according to FDA conversion rules [54], a dose of 3.6 mg/kg of body weight in mice is equivalent to a dose of 0.29 mg/kg in humans. For an adult weighing 70 kg, this equates to 20.6 mg of fullerenes ingested per day. This equates to 25.2 mL (or two-and-a-half 10 mL tablespoons) of an olive oil solution containing 0.8 mg of fullerenes per ml. Since our focus is on the deliberate consumption of the test item by potential human consumers rather than accidental ingestion, it is reasonable to conclude that a dose of 25 mL is significantly higher than the volume that could be consumed daily.

Fullerenes completely dissolved in vegetable oil can cross the intestinal barrier and reach all organs, especially those rich in reticuloendothelial tissue [19]. This has been confirmed by independent research teams in rats [55] and mice [56]. Most importantly, it has also been demonstrated in mice that after administration of ^13^C-enriched C60, a significant proportion can reach the bone within a few hours [57]. These studies demonstrate that the administered C60 undoubtedly reaches the bone marrow. These data clearly validate the results of the current MMT study.

Although required by the official guidelines for validation of the test [49,50], the toxic effect observed in the bone marrow should be interpreted with caution. The decrease in the proportion of PCEs observed with the test substance is more reminiscent of an erythropoietin-like effect than a toxic effect per se [58]. While a decrease in PCEs would necessarily result in anemia, no evidence of anemia or leukopenia was observed in the previous toxicity studies in rats and mice [5,10,19,48].

## 5. Conclusions

Along with the data from the previous acute oral toxicity study, the results of this genotoxicity study should pave the way for further research into fullerenes dissolved in vegetable oils. This could include investigating long-term effects and conducting clinical trials.

However, the history of C60 and in particular the studies carried out by reputable laboratories has reminded us of the importance of properly characterizing the product to be tested before undertaking any toxicity study [7,9]. It is therefore important to specify that this study was carried out on high-purity vacuum-baked fullerenes dissolved in EVOO. It should therefore be remembered that, as with any other drug or drug candidate, the safety of any preparation or new formulation based on C60 intended for human consumption must be demonstrated [4]. Similarly, the quality of the olive oil used to dissolve C60 must be well controlled using an appropriate analytical technique [59].

## Figures and Tables

**Figure 1 nanomaterials-15-00870-f001:**
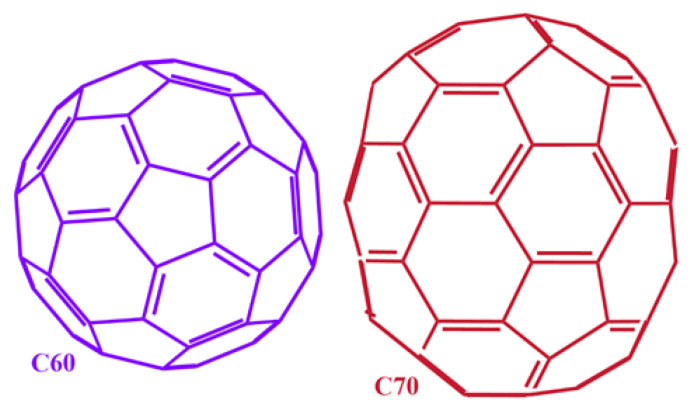
[60]Fullerene or C60 and [70]fullerene or C70. Colours correspond to those of solutions obtained by dissolving each fullerene in a suitable solvent.

**Table 1 nanomaterials-15-00870-t001:** Treatment scheme. NC = negative control. PC = positive control.

Group	Treatment (mg/kg bw)	Dose of Ti (mg/mL)	Total Dose of Fullerenes (mg/kg bw × 2)	N. of Animals	S. Time (h)
NC	Solvent	0.0	0.0	5	24
PC	Cyclophosphamide	6.0	0.0	5	24
Ti	500	50	0.90	5	24
Ti	1000	100	1.80	5	24
Ti	2000	200	3.60	7	24

Ti = Test item, N. = number, S. time = sampling time. The dose of fullerenes is the total dose administered. It was calculated on the basis of the initial concentration of fullerenes in olive oil (0.8 mg/mL) and a density of 893 mg/mL for the test item.

**Table 2 nanomaterials-15-00870-t002:** Mouse micronuclei test data. Sampling time 24 h after the second treatment.

Groups	N. of PCEs Analyzed	MPCEs		PCE/(PCE + NCE)	
		Mean	SD	Mean	SD
Solvent	20,000	5.40	1.14	0.52	0.02
Positive Control	20,000	150.80 */**	9.86	0.37 ** DN	0.02
500 mg/kg bw	20,000	5.00	0.74	0.52	0.01
1000 mg/kg bw	20,000	5.60	1.52	0.49 **	0.01
2000 mg/kg bw	20,000	5.40	1.34	0.46 **	0.01

SD = standard deviation, PCE = polychromatic erythrocyte, NCE = normochromatic erythrocyte, MPCE = number of micronucleated polychromatic erythrocytes relative to 4000PCE. Positive control = 60 mg/kg bw cyclophosphamide. Negative control = Helianthii Annui Oleum Raffinatum. MPCE *: *p* < 0.05 to the vehicle control. MPCE **: *p* < 0.01 to the historical control. PCE/(PCE + NCE) **: *p* < 0.01 to the vehicle and historical control (Kruskal–Wallis non-parametric ANOVA; U: Mann–Whitney U-test versus control; DN = Duncan’s multiple range test).

## Data Availability

Data are contained within the article and Appendix A.

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
