# Peer review of "A Regulatory-Compliant Genotoxicity Study of a Mixture of C60 and C70 Fullerenes Dissolved in Olive Oil Using the Mammalian Micronucleus Test"

_nanomaterials, 2025, doi:10.3390/nano15110870_

Round 1
Reviewer 1 Report
Comments and Suggestions for Authors
The manuscript is well organized and methods used in the experiments are adequate. The manuscript is current and interpreted properly. It can be published after minor revision:
Abstract: First sentence: “… fullerenes — discovered forty (not fourteen!!!) years ago…”
Introduction and Discussion: during discussion of the safety of pristine C60 fullerene [Refs. 10-20], it is necessary to take into account the following important articles:
https://doi.org/10.1080/1536383X.2019.1634055
https://doi.org/10.1007/s12668-020-00762-w
Author Response
The manuscript is well organized and methods used in the experiments are adequate. The manuscript is current and interpreted properly. It can be published after minor revision:
Abstract: First sentence: “… fullerenes — discovered forty (not fourteen!!!) years ago…”
Response: The author would like to sincerely thank the reviewer for the time he devoted to this work and for his valuable comments.
The typo has been corrected in the abstract (highlighted in yellow).
Introduction and Discussion: during discussion of the safety of pristine C60 fullerene [Refs. 10-20], it is necessary to take into account the following important articles:
https://doi.org/10.1080/1536383X.2019.1634055
https://doi.org/10.1007/s12668-020-00762-w
Response: Indeed, both references are worth citing. I have added them and changed the reference numbering accordingly (highlighted in yellow in the revised version).
Reviewer 2 Report
Comments and Suggestions for Authors
This paper by Fathi Moussa presents a rigorous and well-executed in vivo genotoxicity study of a high-purity C60/C70 fullerene mixture, dissolved in extra virgin olive oil, evaluated via the mammalian micronucleus test (MMT) in mice. The study adheres strictly to GLP and OECD/EPA regulatory guidelines and demonstrates the absence of genotoxic effects at the tested doses (which are relevant to potential human consumption). The rationale for selecting a commercially available formulation and the emphasis on regulatory compliance significantly enhance the translational relevance of the findings. I think the experimental design is clear and contributes to the study's robustness. This work addresses a critical gap in the toxicological assessment of fullerene-based consumer products and supports future safety evaluations and potential regulatory acceptance.
I suggest the following optional improvements:
- The authors might consider including biodistribution data or referencing previous studies to reinforce the validity of the negative genotoxic findings.
- A more detailed discussion on the observed erythropoietin-like effect on PCE ratios would add value to the interpretation of bone marrow toxicity indicators.
Author Response
I suggest the following optional improvements:
- The authors might consider including biodistribution data or referencing previous studies to reinforce the validity of the negative genotoxic findings.
- A more detailed discussion on the observed erythropoietin-like effect on PCE ratios would add value to the interpretation of bone marrow toxicity indicators.
Response: The author would like to sincerely thank the reviewer for the time he devoted to this work and for his valuable comments.
References to the biodistribution data and an explanation of the erythropoietin-like effect can be found in the text in the Discussion section. These are references 19, 57-59 and 58, respectively. The reviewer acknowledges that incorporating these data, which are published in table format, into this paper would require permission to be obtained from the editors of each respective journal - a process that could be extremely time-consuming. As this paper is not a general review, I believe the reviewer will agree that including these data as 'optional improvements' is unnecessary, since they have already been published and are accessible to all. However, the author is willing to provide all the relevant documents if necessary. I hope the responses will satisfy the reviewer.
Reviewer 3 Report
Comments and Suggestions for Authors
Author has done a beautiful work on A regulatory compliant genotoxicity study of a mixture of C60 2 and C70 fullerenes dissolved in olive oil using the mammalian micronucleus test.
Introduction
[60]Fullerene or C60 and [70]fullerene or C70. The symbol of [60] and [70] are very similar to reference citation [60] and [70] in Nanomaterials MDPI. Suggest remove [ ]
Line 123-125
Subsection title should be italic 2.1. Experimental design
The sub-subsection titles are normal 2.1.1.Test item and chemicals.
2.1.1.1.Test item
Line 165 'We chose'
Line 92, line 124
Please remove this space line
Regarding space of the each subsection title, the space should be before 12 pt and aft 3 pt.
Line 155
2.1.1.2. should be plain not bold
Line 159
2.1.1.3. should be plain not bold
Same comments as line 201, 212,226,235,262
Line 247-261
Please Consider replace points text to a table
Line 264-290
Please convert the points to paragraphs
line 293-342
Please use justify in paragraph to align the width of text to left margin 4.6 cm and right margin 0.
line 227-335
Line 160 'The other chemicals used'
336-342
table title and description should be in font 9
Discussion
The section is very focus to toxicity of C60 and C70.
It will be great to discuss and compare to other carbon materials. https://doi.org/10.1289/ehp.8266
https://doi.org/10.1183/09031936.06.00071205
10.1016/j.toxlet.2011.05.180
10.1186/1743-8977-10-6
Supplementary material was not shown online. please attach the support information.
Author Response
Author has done a beautiful work on A regulatory compliant genotoxicity study of a mixture of C60 2 and C70 fullerenes dissolved in olive oil using the mammalian micronucleus test.
The author would like to sincerely thank the reviewer for the time he devoted to this work and for his valuable comments.
Introduction
[60]Fullerene or C60 and [70]fullerene or C70. The symbol of [60] and [70] are very similar to reference citation [60] and [70] in Nanomaterials MDPI. Suggest remove [ ]
Response: We have removed the [ ].
Line 123-125
Subsection title should be italic 2.1. Experimental design
The sub-subsection titles are normal 2.1.1.Test item and chemicals.
2.1.1.1.Test item
Response: We have modified them accordingly.
Line 165 'We chose'
Response: The sentence has been changed.
Line 92, line 124
Please remove this space line
Regarding space of the each subsection title, the space should be before 12 pt and aft 3 pt.
Response: We have modified the spaces accordingly.
Line 155
2.1.1.2. should be plain not bold
Response: It has been done
Line 159
2.1.1.3. should be plain not bold
Same comments as line 201, 212,226,235,262
Response: It has been done
Line 247-261
Please Consider replace points text to a table
Response: As replacing the text in lines 247–261 with a table would be difficult, we have converted it into paragraphs instead, as the reviewer suggested for lines 264–290.
Line 264-290
Please convert the points to paragraphs
Response: We have converted them accordingly.
line 293-342
Please use justify in paragraph to align the width of text to left margin 4.6 cm and right margin 0.
line 227-335
Response: It has been done.
Line 160 'The other chemicals used'
Response: This paragraph has been modified.
336-342
table title and description should be in font 9
Response: It has been done.
Discussion
The section is very focus to toxicity of C60 and C70.
It will be great to discuss and compare to other carbon materials. https://doi.org/10.1289/ehp.8266
https://doi.org/10.1183/09031936.06.00071205
10.1016/j.toxlet.2011.05.180
10.1186/1743-8977-10-6
Response: Yes, the reviewer is right, "the section is very focus to toxicity of C60 and C70". However, we believe that the reviewer will agree that the aim of this study, which was conducted in an accredited GLP laboratory in accordance with FDA and OECD guidelines, was to assess the genotoxicity of fullerene solutions that are sold to consumers online. Therefore, we consider it off-topic to discuss or compare this study with other academic research on carbon nanomaterials. However, I agree that the reviewer's idea could form the basis of a very interesting general review of the subject.
Supplementary material was not shown online. please attach the support information.
Response: The supplementary material has been attached.
Round 2
Reviewer 3 Report
Comments and Suggestions for Authors
Author has improved the manuscript.
There are some minor corrections.
Line 244-255
The new paragraphs should be consist with third person passive tense.
Line 256 Sub section title should be Italic
2.3. Statistics and evaluation of experimental data and interpretation.
Line 265-279
The new paragraphs should be consist with third person passive tense.
Line 305
Please use no bold italic font for
3.3. Frequency of micronuclei.
Please remove line 330 and 332 spaces
same apply for 388 and 390
line 405
https://www.mdpi.com/article/doi/s1
Please
Author Response
There are some minor corrections.
Line 244-255
The new paragraphs should be consist with third person passive tense.
Response: The paragraphs have been rewritten as requested.
Line 256 Sub section title should be Italic
Response: it has been done.
2.3. Statistics and evaluation of experimental data and interpretation.
Line 265-279
The new paragraphs should be consist with third person passive tense.
Response: The paragraphs have been rewritten as requested.
Line 305
Please use no bold italic font for
3.3. Frequency of micronuclei.
Response: it has been done
Please remove line 330 and 332 spaces
same apply for 388 and 390
Response: it has been done
line 405
https://www.mdpi.com/article/doi/s1
Please
Response: it has been done